# Age-specific biological and molecular profiling distinguishes paediatric from adult acute myeloid leukaemias

Shahzya Chaudhury[1,2], Caitríona O'Connor[1], Ana Cañete[1], Joana Bittencourt-Silvestre [1], Evgenia Sarrou [1], Áine Prendergast[1], Jarny Choi [3], Pamela Johnston[4], Christine A. Wells [3], Brenda Gibson[2] & Karen Keeshan [1]

Acute myeloid leukaemia (AML) affects children and adults of all ages. AML remains one of the major causes of death in children with cancer and for children with AML relapse is the most common cause of death. Here, by modelling AML in vivo we demonstrate that AML is discriminated by the age of the cell of origin. Young cells give rise to myeloid, lymphoid or mixed phenotype acute leukaemia, whereas adult cells give rise exclusively to AML, with a shorter latency. Unlike adult, young AML cells do not remodel the bone marrow stroma. Transcriptional analysis distinguishes young AML by the upregulation of immune pathways. Analysis of human paediatric AML samples recapitulates a paediatric immune cell interaction gene signature, highlighting two genes, RGS10 and FAM26F as prognostically significant. This work advances our understanding of paediatric AML biology, and provides murine models that offer the potential for developing paediatric specific therapeutic strategies.

[1] Paul O'Gorman Leukaemia Research Centre, Institute of Cancer Sciences, University of Glasgow, Glasgow, UK. [2] Royal Hospital for Children, Glasgow, Scotland, UK. [3] Centre for Stem Cell Systems, Faculty of Medicine, Dentistry and Health Sciences, The University of Melbourne, Melbourne, Australia. [4] School of Veterinary Medicine, College of Medical, Veterinary and Life Sciences, University of Glasgow, Glasgow, UK. These authors contributed equally: Shahzya Chaudhury, Caitríona O'Connor. Correspondence and requests for materials should be addressed to K.K. (email: karen.keeshan@glasgow.ac.uk)

The incidence of acute myeloid leukaemia (AML) increases with age, and in childhood accounts for 20% of all leukaemia. The current overall survival rate in children is only 60–75%, and thereafter falls progressively with age to 5–15% in the elderly. Both children and adults die from a combination of relapse (up to 35% and 99%, respectively) and treatment-related mortality during both induction and consolidation therapy[1,2]. AML is characterised by impaired myeloid differentiation resulting in the accumulation of myeloid blasts in the bone marrow (BM) and peripheral blood (PB). Seminal studies in adult AML[3] have led to the leukaemia stem cell (LSC) hypothesis, which postulates that leukaemias are organised into cellular hierarchies, mirroring normal haemopoiesis. LSCs have similar properties to normal adult HSCs at the apex of the haemopoietic hierarchy, which differentiate into bulk leukaemia cells. In the majority of adult human AMLs, the LSC has been identified as either the LSK or a more mature progenitor cell that has acquired self-renewal[4,5]. Current therapies fail to eradicate leukaemic cells, which are protected in the BM microenvironment, interact with the surrounding cells, and cause disease relapse[6,7].

There are major differences between paediatric and adult AML relating to (i) the frequency of de novo AML versus secondary AML subsequent to underlying myeloproliferative neoplasms (MPN) or myelodysplastic syndromes (MDS) and (ii) cytogenetic and molecular abnormalities[8–10]. In children, the vast majority of patients present with de novo AML while in adults, a significant proportion of AML arises from an underlying MPN or MDS and this characteristically increases with age. This can be explained by the significant differences in genetic landscapes of paediatric and adult AML. Only 20% of paediatric patients have a normal karyotype and the number of somatic mutations is lower than in adult AML (5 per paediatric sample versus 10–13 per adult sample). Paediatric AML has a higher frequency of cytogenetic abnormalities compared to adult, with some occurring almost exclusively in infants/children. Furthermore, the epigenetic landscapes of paediatric and adult AML are vastly different with respect to the incidence and type of mutations in epigenetic modulators[11]. The recent TARGET AML initiative comprehensively showed the similarities and differences in the mutational profile of >1000 AML patients across the age spectrum, demonstrating DNA methylation and miRNA profiles can stratify paediatric patients in terms of overall and progression-free survival, calling for an update to address-specific vulnerabilities of paediatric subtypes[12]. To progress our understanding of paediatric AML, it is important to establish models of disease that recapitulate features of the disease to develop age-specific therapies.

Aging features, such as decreased immune response, increased myeloid lineage skewing, genomic instability, reduced regenerative capacity of stem cells, and positive selection of pre-leukaemic clones driven by cell intrinsic and extrinsic factors, are all thought to contribute to the high incidence of AML in older adults[1,13]. This does not explain the occurrence of AML in children. Neonatal HSCs are a cycling population compared to the quiescent properties of adult HSCs. With respect to the infant haemopoietic system (where the incidence of childhood AML is at its highest), children up to the age of three retain features of foetal haemopoiesis. The equivalent switch from foetal to adult HSCs (quiescent cells) in mice occurs between 3 and 4 weeks of age[14,15]. Aged HSCs and the surrounding stromal cells possess differences in lineage output, and have altered genetic profiles (e.g. inflammatory, DNA damage) compared to their younger adult counterparts[16,17]. BM stromal cells including mesenchymal stromal cells (MSCs), osteoblasts (OBs), and endothelial cells (ECs) contribute to HSC function. Similar to HSCs, leukaemic cells depend on the BM niche, and not only co-opt normal niches but also alter them to their own advantage for propagation and

treatment survival[18–20]. It is not known whether changes that occur in the leukaemic BM niche are the same in paediatric and adult AML.

Given the large spectrum of genetic abnormalities found in AML, there are many examples that have been shown to recapitulate AML disease in the mouse, such as MLL and PML oncofusions, and those that present pre-leukaemic features of the disease, but not overt AML, such as AML1ETO[21–23]. NUP98HOXA9 (NH9) oncogene represents 11p rearrangements found in both paediatric and adult AML, and it has been exclusively associated with myeloid leukaemia both in humans and in experimental models. Although 11p rearrangements are rare, NH9 is representative of deregulated HOX leukaemia, observed in 70% of all AMLs as a result of several upstream genetic alterations that occur across all age-groups[24,25]. We have modelled the age continuum using the NH9 oncogene to address how the age of the cell of origin impacts on leukaemic disease. We show that paediatric AML is biologically different to adult AML, and that the age of the cell of origin significantly influences leukaemia latency, lineage, molecular profile, and the leukaemic niche.

## Results

**Comparable in vitro transformation of LSKs from all ages.** In order to model the continuum of leukaemia based on the age relationship estimated between humans and mice[26], we isolated murine lin⁻sca1⁺cKit⁺ (LSK) cells, which include the long- and short-term HSCs, and myeloid progenitor populations (CMPs and GMPs) from different ages of mice (prenatal and postnatal) to represent infant/prenatal origins (foetal liver (FL)), childhood (BM from 3 week(w)), young adult (10 w) and middle/old age (>52w) C57BL/6 mice. These different aged populations were transduced with NH9 retrovirus and assessed for in vitro transformation by CFC assay. After the first plating (P1) colonies were enumerated and identified (Supplementary Figure 1a), and replated up to three times (P2 and P3) (Fig. 1a). The ability of cells to produce colonies at P3 indicates the acquisition of self-renewal capacity and suggests in vitro transformation[27]. LSKs transduced with empty vector control MigR1 did not give rise to any colonies past P1 (Supplementary Figure 1b). Analysis of cells from LSK-derived (P3) colonies showed no difference in the median fluorescent intensity (MFI) of GFP between the age groups, indicating similar expression of NH9 (Fig. 1b). Expression of the NH9 fusion gene was also detected in NH9 transduced cells from LSK-derived (P1 and P3) colonies by RT-PCR (Fig. 1c). NH9 transduced LSKs efficiently produced large numbers of colonies by P3 independent of age (Fig. 1d). However far fewer colonies were produced by NH9 transduced FL CMPs compared to other ages, whereas NH9 transduced FL GMPs failed to replate to P3 (Fig. 1e, f, Supplementary Figure 1c, d). We next examined the intrinsic transformability of foetal LSKs, CMPs and GMPs with common AML drivers, AML1ETO and FLT3-ITD as well as NH9. Only LSKs from FL expanded in OP9 co-culture with different oncogenes, while FL CMPs and GMPs exhausted in long-term culture suggesting that there is an intrinsic feature of foetal committed myeloid progenitors that inhibits leukaemic transformation (Supplementary Figure 1e–g). Immunophenotypic analysis of cells from P2 showed the absence of lymphoid markers and expression of myeloid markers with a significant difference in the expression of Gr1 in FL compared to 10w and >52 W NH9 transduced LSKs (Supplementary Figure 1h). By P3 colony morphology did not differ greatly between ages of NH9 transduced LSKs. Each age population gave rise to granulocyte erythroid macrophage megakaryocyte (GEMM), granulocyte macrophage (GM), granulocyte (G), macrophage (M) and erythroid (E) colonies, although there was a significantly higher

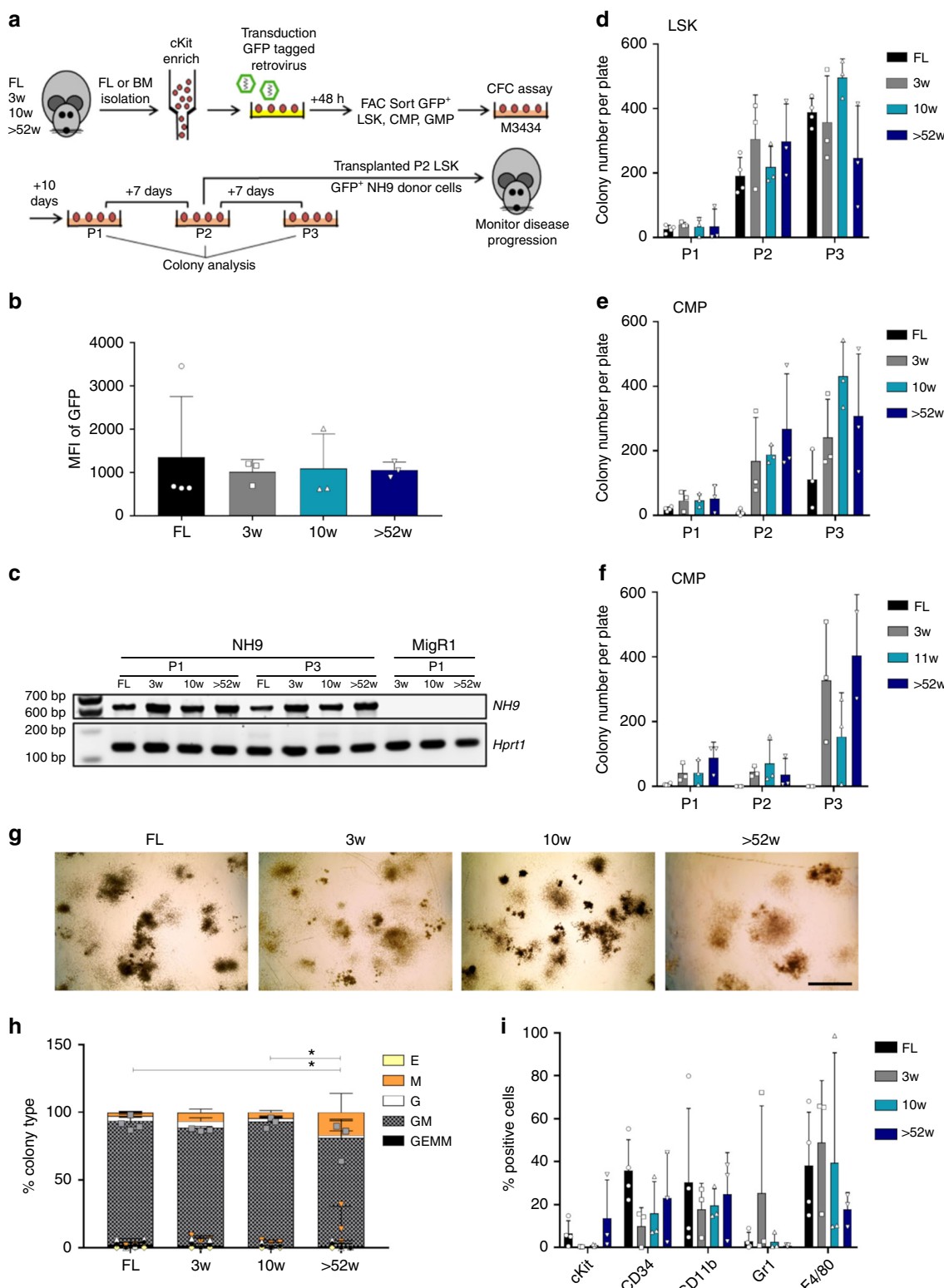

proportion of mature colonies derived from >52w LSKs compared with colonies derived from FL and 10w LSKs (Fig. 1g, h). Immunophenotypic analysis showed no significant difference in the expression of mature myeloid (CD11b, Gr1 and F4/80) or stem cell (cKit and CD34) markers between the ages (Fig. 1i). These results showed that foetal transformation was restricted to the immature LSK compartment and LSKs from all ages transformed comparably in vitro.

**Different leukaemia latencies due to age of cell of origin**. To investigate the influence of the BM niche on age-defined LSK homing potential, NH9-transduced P2 LSKs from each group were transplanted into recipient mice and assessed after 18 h for homing to different organs (Fig. 2a). NH9 transduced cells of 10W and >52W origin homed significantly more to the BM and spleen compared to FL and 3W NH9 transduced cells (FL vs >52w $p = 0.0000006$ (BM) $p = 0.02$ (spleen), FL vs 10w $p = 0.015$

**Fig. 1** LSKs from all ages display comparable in vitro transformation ability. **a** Schematic of CFC assay and transplantation model. Murine FL or BM cells were isolated from mice of ages indicated. cKit enriched cells were transduced with GFP tagged NH9 virus. After 48 h the cells were sorted for GFP$^+$ LSKs, CMPs and GMPs and plated in methylcellulose media (M3434). Cells were replated to P3 and colonies were counted and analysed at each plating. Cells from P2 colonies were transplanted into sub-lethally irradiated recipient mice. **b** Graph of cells from P3 showing no difference in the MFI of GFP with cellular age. **c** Gene expression by RT-PCR of NH9 and Hprt1 (endogenous control) in NH9 (FL, 3w, 10w, >52w) or MigR1 (3w, 10w, >52w) transduced LSKs from P1 or P3 where indicated. Graphs of the number of colonies per plate from serial replating CFC assay, produced by NH9 transduced **d** LSKs, **e** CMPs and **f** GMPs. **g** Representative morphology of colonies at ×2 magnification from each age group at P3. Black scale bar represents 2 mm. **h** Graph of percentage colony type produced at P3 by NH9 transduced LSKs. GEMM Granulocyte erythroid macrophage megakaryocyte, GM Granulocyte macrophage, G granulocyte, M macrophage and E erythroid. **i** Graph of surface marker expression at P3 showing no significant differences with cellular age. **b, d–f, h, i**) Bars depict mean + s.d. Significance determined by one-way ANOVA and Bonferroni post-test. FL $n = 4$, 3w $n = 3$, 10w $n = 3$, > 52w $n = 3$, *$p < 0.05$

(BM) $p = 0.014$ (spleen), 3w vs >52w $p = 0.00003$ (BM), 3w vs 10w $p = 0.006$ (BM) $p = 0.004$ (spleen). To investigate the influence of the BM niche on age-defined LSK transformability, NH9-transduced P2 LSKs from each age group were transplanted into recipient mice generating NH9$^{FL}$, NH9$^{3W}$, NH9$^{10W}$ and NH9$^{>52W}$. No significant differences in long-term engraftment, determined by GFP expression in PB after 16w, were observed across the groups (Fig. 2b). Recipients were monitored for clinical signs of leukaemia and sacrificed upon leukaemia development, or after 1 year if no leukaemia had developed. Mice transplanted with adult cells (NH9$^{10W}$ and NH9$^{>52W}$) developed leukaemia with a significantly shorter latency compared to young leukae-mias (NH9$^{FL}$ and NH9$^{3W}$) ($p = 0.0111$) (Fig. 2c). Pooling the results into young (NH9$^{FL}$ and NH9$^{3W}$) and adult (NH9$^{10W}$ and NH9$^{>52W}$) revealed a 70 day increase in median survival (MS) in the young compared to adult groups (MS 315 days vs 245 days, respectively) (Fig. 2d). Furthermore, while 100% of mice trans-planted with adult NH9$^+$ LSKs developed disease, there was a significantly lower disease penetrance in mice receiving young NH9$^+$ LSKs (50–75%) (Figs. 2h, 3b). There were no age-related differences in disease burden in leukaemic mice as assessed by BM and spleen cellularity, and spleen weight (Fig. 2e). Similarly, leukaemic mice from all groups displayed equivalent levels of leukocytosis, thrombocytopenia and anaemia (Fig. 2f). There were equivalent percentages of GFP$^+$ cells in the PB, BM and spleen of leukaemic mice at the time of death confirming equivalent disease burden in all groups (Fig. 2g). In addition, there was no difference in the expression of the stem cell markers CD34 and cKit on the leukaemic cells derived from young or adult NH9$^+$ LSKs (Fig. 2i). These data show that the age of the cell of origin significantly impacts disease latency.

**Different lineage phenotypes due to age of cell of origin**. We next performed in-depth phenotypic analysis of the leukaemic blasts by flow cytometry and immunohistochemistry (IHC). Using markers of myeloid and lymphoid leukaemia, we identified three distinct types of acute leukaemia driven by NH9. All mice transplanted with adult NH9$^+$ LSKs developed AML (Fig. 3a, b) characterised by intermediate CD11b and Gr-1 expression levels (i.e. overt AML distinct to myeloproliferative phenotype which was observed prior to clinical signs of AML and characterized by CD11b$^{hi}$Gr-1$^{hi}$ expression), with low or no expression of lym-phoid markers CD19, B220, CD4 and CD8 (Fig. 3a, c) and positive staining for MPO by IHC (Fig. 3d). Mice transplanted with young NH9$^+$ LSKs developed AML but also developed two other disease phenotypes in vivo; acute lymphoblastic leukaemia (ALL) and mixed phenotype acute leukaemia (MPAL) (Fig. 3a, b). The ALL disease was characterised by the absence of myeloid markers CD11b and Gr-1, and the expression of high levels of either CD19 or B220 on leukaemic blasts (Fig. 3a, c) as well as positive staining for CD3 by IHC (Fig. 3d). The MPAL disease expressed myeloid markers with low or no expression of

lymphoid markers by flow cytometry (Fig. 3a, c), but positive for both MPO and CD3 intracellularly by IHC (Fig. 3d)—an estab-lished diagnostic criteria for MPAL[28]. Despite the development of lymphoid disease in vivo, the P2 cells did not show any lymphoid cell surface marker expression or lymphoid potential in vitro prior to transplantation (Supplementary Figure 1h, i). When we transplanted healthy LSKs from FL (LSK$^{FL}$) and >52w BM (LSK$^{>52W}$) into 6–8w recipients we saw no age-dependent skewing of donor cells, with both LSK$^{FL}$ and LSK$^{>52w}$ groups having the same proportion of myeloid, B and T cells (Supple-mentary Figure 1j, k). Therefore, the disease lineage skewing in vivo is a result of NH9 expression specifically in young LSKs.

We characterized the three distinct disease phenotypes arising from the young NH9$^+$ LSKs by gene expression analysis of lineage restricted genes. In the GFP$^+$ leukaemic blasts from the AML diseases there was upregulation of myeloid genes Cebpb, Cebpe and Fcgr3 compared with ALL blasts (Fig. 3e), and lower expression of these genes in the MPAL cells compared with the AML cells. We observed higher expression of key B lineage genes Cd79a, Ptprc, Pou2af1, Foxo1 and Vpreb1 in the ALL cells compared with AML or MPAL derived cells (Fig. 3f). In addition, expression of lymphoid genes associated with the T-cell lineage Rag1, Rag2, Il7ra and Flt3 were also elevated in ALL blasts when compared to AML and MPAL leukaemic cells (Fig. 3g). Together, our in vivo results demonstrate that upon transformation the age of the cell of origin results in biologically different diseases suggesting that young LSKs have strong intrinsic lymphoid programmes.

**Differential regulation of the BM niche by 3w and >52w AML**. Our gene expression analysis of the three disease phenotypes demonstrated lineage gene expression changes, but whether the age of the cell of origin impacts the transcriptional profile of the myeloid leukaemias which developed from all age groups was unknown. Thus, RNA-seq was performed on in vivo generated myeloid leukaemias (NH9$^{FL}$, NH9$^{3W}$, NH9$^{10W}$ and NH9$^{>52W}$) (Fig. 4a). Principal component analysis (PCA) of our RNA seq data revealed that samples clustered together according to age (Fig. 4b). Significant differentially expressed genes (DEGs) were identified in pairwise comparisons between NH9$^{FL}$ and each age group and filtered for log fold change (LFC) of ≥1.5 and $q$-value of <0.05. In this comparison, we identified 795 DEGs in NH9$^{3W}$ (537 upregulated in NH9$^{FL}$, 258 upregulated in NH9$^{3W}$), 2117 DEGs in NH9$^{10W}$ (1475 upregulated in NH9$^{FL}$, 642 upregulated in NH9$^{10W}$) and 1455 DEGs in NH9$^{>52W}$ (501 upregulated in NH9$^{FL}$, 953 upregulated in NH9$^{>52W}$) (Supplementary Fig-ure 2a). Therefore despite all samples representing GFP$^+$ NH9-driven myeloid leukaemic blasts, the transcriptional profiles were substantially different according to the age of the cell of origin. Overlapping those genes upregulated in NH9$^{3W}$, NH9$^{10W}$ and NH9$^{>52W}$ cells in comparison to NH9$^{FL}$ (Supplementary Fig-ure 2b) identified a number of genes, which were uniquely

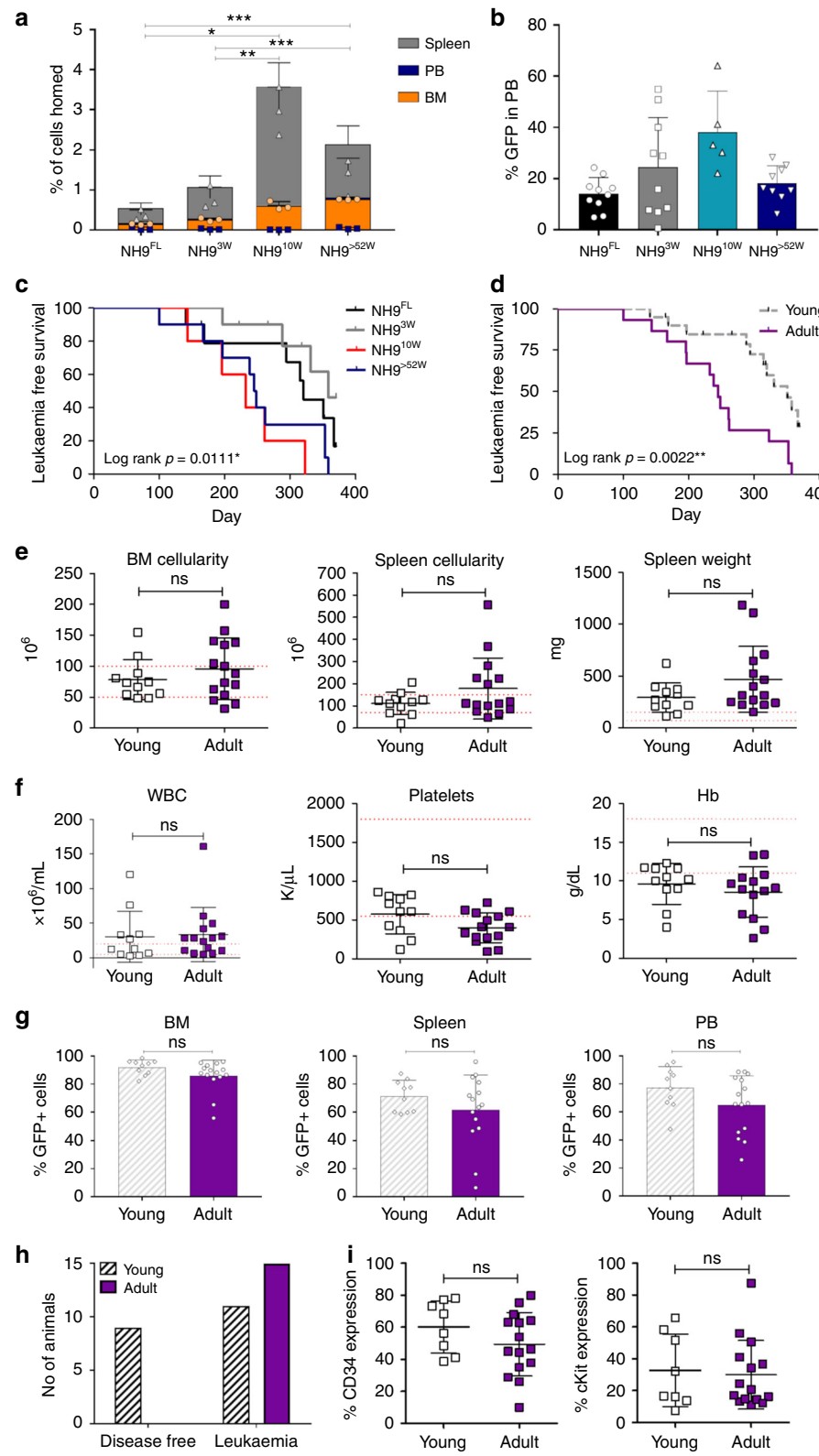

upregulated in each age group compared with NH9$^{FL}$. These sets of genes were analysed for gene ontology (GO) pathway enrichment using the MSigDB platform. The top ten most significantly enriched GO pathways revealed an enriched development signature in the NH9$^{3W}$ leukaemic cells (Supplementary Figure 2c),

enriched immune response in NH9$^{10W}$ leukaemic cells (Supplementary Figure 2d) and enrichment of cell–cell interaction pathways in NH9$^{>52W}$ leukaemic cells (Supplementary Figure 2e).

The in vivo differences in disease latency and phenotype together with the transcriptional analysis (Supplementary

**Fig. 2** Leukaemia latencies differ with different ages of cells of origin. **a** Graph of percentage of cells that homed to BM, PB and spleen 18 h post transplant of NH9 LSK$^{FL}$ ($n = 3$), LSK$^{3w}$ ($n = 3$), LSK$^{10w}$ ($n = 3$) and LSK$^{>52w}$ ($n = 3$). Significance determined by Student's $t$-test, $*p < 0.05$, $**p < 0.01$, $***p < 0.001$. **b** Percentage of GFP$^+$ cells in PB at 16w. Graph depicts mean + s.d. from two independent experiments. Significance determined by one-way ANOVA and Bonferroni post test ($p = 0.44$). **c** Kaplan–Meier survival curve of recipients transplanted with NH9 transduced LSKs from P2. Recipients censored at death. Significance determined by overall log rank, $*p < 0.05$. **d** Kaplan–Meier survival curve of recipients pooled into young (NH9$^{FL}$ and NH9$^{3W}$) and adult (NH9$^{10W}$ and NH9$^{>52W}$). Recipients censored at death. Significance determined by overall log rank, $**p < 0.01$. **e** Leukaemic burden assessed by BM and spleen cellularity, and spleen weight. **f** Analysis of PB parameters of recipients with donor derived leukaemia. **e**, **f** Graphs show individual results with mean ± s.d. Red dotted lines depict normal range. Young $n = 11$, adult $n = 15$. ns not significant as determined by Student's $t$-test. **g** Assessment of leukaemic infiltration as %GFP in BM, spleen and PB after recipient mice developed donor derived leukaemia, from young and adult groups. Young $n = 10$, adult $n = 15$. Graphs depict mean + s.d. ns not significant as determined by Student's $t$-test. **h** Graph of the total number of mice from two independent experiments developing donor-derived leukaemia in young and adult groups. **i** Expression of immature surface markers on GFP$^+$ cells from BM of recipients that succumbed to NH9 AML. Graphs show individual result with mean ± s.d. Young $n = 8$, adult $n = 15$. ns not significant as determined by Student's $t$-test. All data representative of two independent experiments. NH9$^{FL}$ $n = 10$, NH9$^{3w}$ $n = 10$, NH9$^{10w}$ $n = 5$, NH9$^{>52w}$ $n = 10$, young $n = 20$ and adult $n = 15$ unless otherwise stated

---

Figure 2c–e) suggests differences in niche cell interactions across the age groups. Thus we next characterized the effect of the age of a leukaemic cell on the BM niche by performing secondary BM transplantations with leukaemic cells of the same phenotype (AMLs) from NH9$^{3W}$ and NH9$^{>52W}$ mice—as there was the greatest discrimination between those groups by PCA (Fig. 4b). Using an established flow cytometry approach for identifying BM stromal cells (Fig. 4c) we identified ECs (CD45$^-$Ter119$^-$CD31$^+$), MSCs (CD45$^-$Ter119$^-$CD31$^-$CD51$^+$ PDGFRa$^+$) and OBs (CD45$^-$Ter119$^-$CD31$^-$CD51$^+$PDGFRa$^-$) in the central marrow or compact bone (collagenase-treated bone fragments). We assessed the disease burden of transplanted mice with >30% GFP expression in the BM (Fig. 4d, individual mice are indicated on the graphs in Fig. 4d–h with different symbols) by peripheral white blood cell (WBC) counts (Fig. 4e) at the time of sacrifice, and demonstrated no differences between the two cohorts. There was a significant increase in the percentage of ECs in the central marrow of mice transplanted with NH9$^{>52W}$ cells (Fig. 4f) with no changes in the percentage of MSCs compared to control non-transplanted (NT) mice and mice transplanted with NH9$^{3W}$ leukaemic cells. There was also a significant increase in the percentage of MSCs and OBs in the compact bone of mice transplanted with NH9$^{>52W}$ cells compared to NH9$^{3W}$ (Fig. 4h). These changes are specific to the leukaemic niche as the effect on ECs, MSCs and OBs was not evident in the BM stromal cells of mice that were transplanted with normal LSKs of foetal (LSK$^{FL}$) or >52w (LSK$^{>52W}$) origin (Fig. 4g, i). There was a small decrease in the percentage of ECs in the central marrow of mice receiving LSK$^{>52W}$ compared to the LSK$^{FL}$ group and NT mice. No differences in ECs, MSCs or OBs were seen in mice transplanted with NH9$^{3W}$ leukaemic cells (Fig. 4f, h), with their stromal compartment similar to a NT mouse. Our data show significant remodelling of the BM niche in mice transplanted with adult leukaemia cells that is not evident when the leukaemia has developed from a young cell of origin. These data indicate that the leukaemic niche is not the same when the leukaemia has developed from a young or adult cell of origin.

**A distinct molecular profile identified for paediatric AML**. The differences in the leukaemic niche of NH9$^{3W}$ transplanted mice compared to NH9$^{>52W}$ may be explained by differential inter-actions between the young leukaemic cells and niche cells as a result of their intrinsic molecular profile. To test this, we com-pared the transcriptional profile of the in vivo generated NH9$^{3W}$ and NH9$^{>52W}$ myeloid leukaemias. To control for genes differ-entially expressed with normal aging, RNA-seq was performed on LSKs of 3w (LSK$^{3W}$) or >52w (LSK$^{>52W}$) origin (Supplementary Figure 3a). Comparing NH9$^{3W}$ and NH9$^{>52W}$ myeloid leukaemias

using a cutoff of LFC ≥1.5 and $q$-value of <0.05, a total of 382 DEGs were identified (Fig. 5a)—164 upregulated in NH9$^{3W}$ and 218 upregulated in NH9$^{>52W}$ leukaemic cells. Comparing LSK$^{3W}$ with LSK$^{>52W}$ a total of 1038 DEGs were identified (Fig. 5b)—236 upregulated in LSK$^{3W}$ and 802 upregulated in LSK$^{>52W}$ leu-kaemic cells. Only 7 genes were upregulated in both LSK$^{3W}$ and NH9$^{3W}$ DEGs, with 21 upregulated genes in common between the LSK$^{>52W}$ DEGs and NH9$^{>52W}$ DEGs (Fig. 5c, Supplementary Data 1, 2). All further analysis of DEGs excludes these genes so all results indicate age-specific leukaemia changes rather than changes due to healthy age. Our sets of DEGs (Supplementary Data 3, 6) were then interrogated for GO pathway enrichment using the MSigDB platform. In NH9$^{3W}$ cells the ten most sig-nificantly enriched GO pathways included pathways related with the immune system and response to oxygen (Fig. 5d, Supple-mentary Data 7). There were ten genes common in five or more of these top ten pathways (Fig. 5e), which included two innate immune cell genes involved in toll-like receptor (TLR) signalling —Cd180 and Ticam2. Hierarchical clustering analysis of one hundred TLR associated genes (KEGG) revealed enrichment of this pathway in NH9$^{3W}$ compared to NH9$^{>52W}$ leukaemic cells (Supplementary Figure 3c). In contrast, in NH9$^{>52W}$ leukaemic cells compared to NH9$^{3W}$, the ten most significantly enriched GO pathways included pathways mostly related with metabolic pro-cesses, and response to endogenous stimulus (Supplementary Figure 3b, Supplementary Data 8). Pathways related with tissue development and adhesion were enriched in cells of both LSK$^{3w}$ and LSK$^{>52w}$ origin (Supplementary Figure 3d, e, Supplementary Data 9, 10). LSK$^{3W}$ also showed enrichment of cell proliferation and immune system pathways, while cell differentiation pathways were enriched in LSK$^{>52W}$.

To validate our findings that leukaemic cells of 3w and >52w origin interact with the BM niche in different ways, a cancer-stroma interaction database containing 628 validated ligand–receptor interactions was used[29]. These include multiple signals such as cytokine and chemokine receptors, adhesion molecules, Notch, Wnt, and hedgehog pathways—key regulators of HSC/LSC interactions in the BM niche. Hierarchical clustering analysis of receptors expressed in the NH9$^{3W}$ and NH9$^{>52W}$ leukaemic cells was performed using the Stemformatics plat-form[30] (Supplementary Figure 3f). This analysis confirmed differential expression of receptors between NH9$^{3W}$ and NH9$^{>52W}$ leukaemic cells, in two-specific clusters annotated (i) and (ii) (Supplementary Figure 3f, g). Two genes from cluster (ii) were significantly upregulated in NH9$^{3W}$ leukaemic cells (Fig. 5f) including Tnfrsf13b, a TNF receptor, found primarily on B cells, involved in B-cell signalling and survival[31]. Analysis of DEGs upregulated in NH9$^{3W}$ myeloid blasts using STRING[32] revealed a

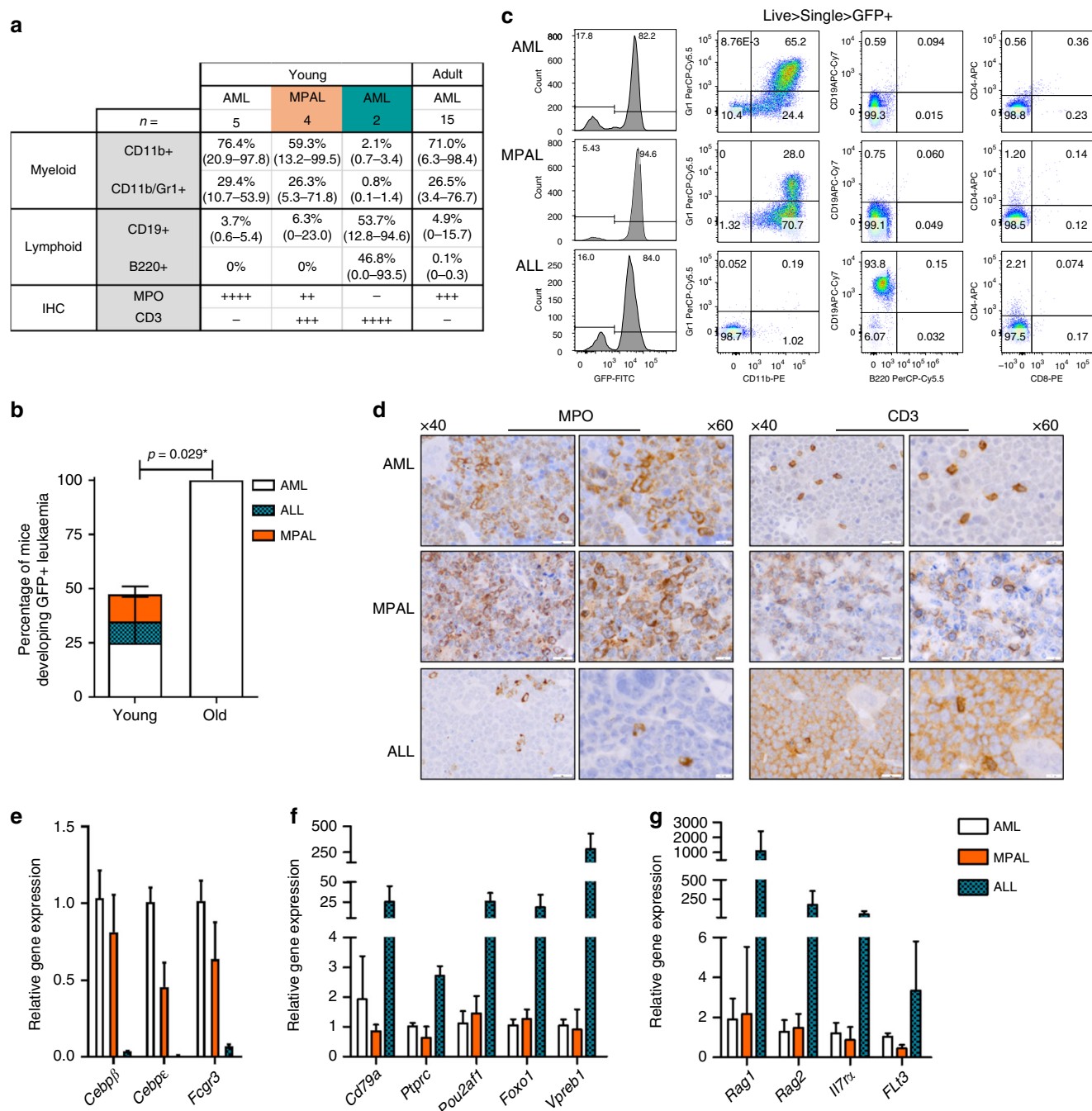

**Fig. 3** Different leukaemia lineage phenotypes due to age of cell of origin. **a** Summary for the outcome for young or adult mice developing NH9+ leukaemia. Average expression of CD11b, CD11b/Gr-1, CD19 and B220 in GFP+ BM cells is shown with expression range. For IHC '−' denotes neoplastic cells are negative for the marker, '+' denotes 20%, '++' 20–40%, '+++' 40–60% and '++++' more than 60% positive expression in neoplastic cells. **b** Bar chart of lineage of all donor derived leukaemia showing data pooled into young (NH9FL and NH93w) (n = 10) and adult (NH910w and NH9>52w) (n = 15). Graphs depict mean + s.d. from two independent experiments. Significance determined by two-way ANOVA, *p < 0.05. **c** Representative flow cytometry plots from each disease phenotype AML (top row), MPAL (middle) and ALL (bottom). Flow cytometry plots depict CD11b, Gr-1, B220, CD19, CD8 and CD4 expression through GFP in the BM. **d** Representative ×40 and ×60 IHC images of sternal sections from each disease phenotype AML (top row), MPAL (middle) and ALL (bottom). Left two columns shows MPO and right 2 columns shows CD3. Positive cells exhibit brown cytoplasmic staining. Black scale bar represents 20 μm for ×40 images and 500 μm for ×60 images. Relative gene expression of **e** myeloid **f** B-lymphoid and **g** T-lymphoid genes in AML (n = 3), MPAL (n = 4) and ALL (n = 2) generated in vivo from young LSKs transformed with NH9. Graphs depict mean LFC + s.d

hub of protein–protein interactions surrounding Cd180 (Fig. 5g). These genes included innate immune cell related genes Ly86, Cd48 and Fam26f. We next analysed the NH93W DEGs in human AML samples using two separate datasets with either paediatric (GSE17855, n = 237) or adult (GSE6891, n = 537) AML samples

that had been run on the same microarray platform. We corrected for batch differences using reference genes with low variance and mean (see Methods). From our DEG list we identified 19 genes which were significantly enriched in human paediatric AML samples relative to adult AML (Fig. 5h). This

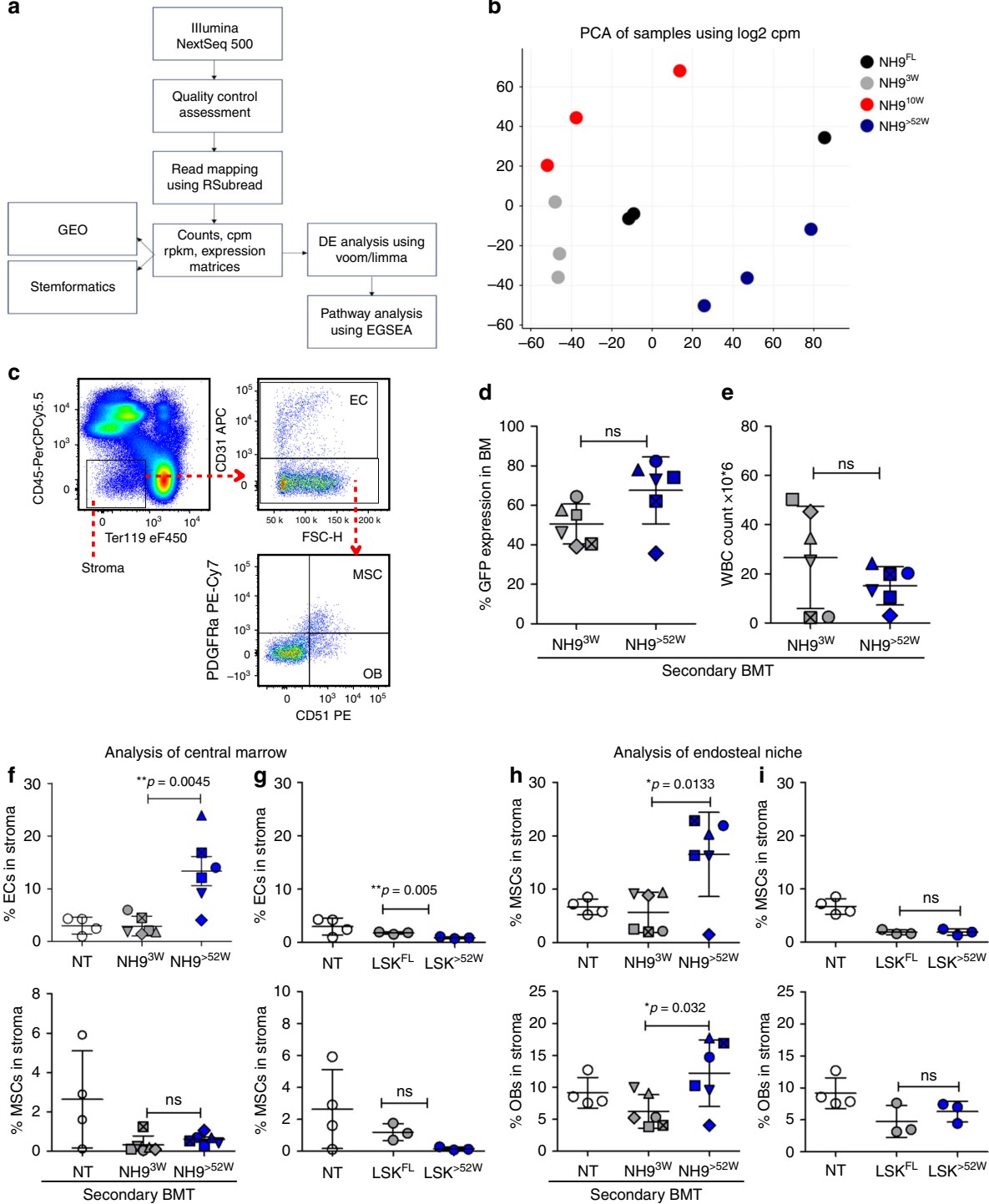

**Fig. 4** Differential regulation of the BM niche by >52w and 3w AML. **a** Schematic overview of RNA seq data acquisition and analysis. **b** Plot of principal component analysis of gene expression in NH9[FL], NH9[3w], NH9[10w] and NH9[>52w] myeloid leukaemias, $n = 3$ in each group. **c** Flow cytometric gating strategy used to identify endothelial cells (EC), mesenchymal cells (MSC) and osteoblasts (OB) in the stroma (CD45−Ter119−). Leukaemic burden assessed by **d** %GFP in BM and **e** WBC count. Analysis of percentage ECs (top) and MSCs (bottom) in the stroma of central marrow of mice transplanted with **f** NH9[+] AML blasts or **g** healthy LSKs of foetal (LSK[FL]) or >52w (LSK[>52W]) origin. Analysis of percentage OBs (bottom) and MSC (top) in the stroma of the compact bone of mice transplanted with **h** NH9[+] AML blasts or **i** LSK[FL] or LSK[>52W]. **d**–**i** Graphs depict individual values with mean ± s.d. of mice transplanted with leukaemic blasts from NH9[3W] ($n = 13$) or NH9[>52W] ($n = 9$) (only mice with >30% GFP in the bone marrow are included, and individual mice are indicated with different symbols on the graphs), and LSK[FL]($n = 4$) or LSK[>52w]($n = 4$) from healthy mice. NT not transplanted ($n = 4$). Significance determined by Student's $t$-test, *$p < 0.05$, **$p < 0.01$

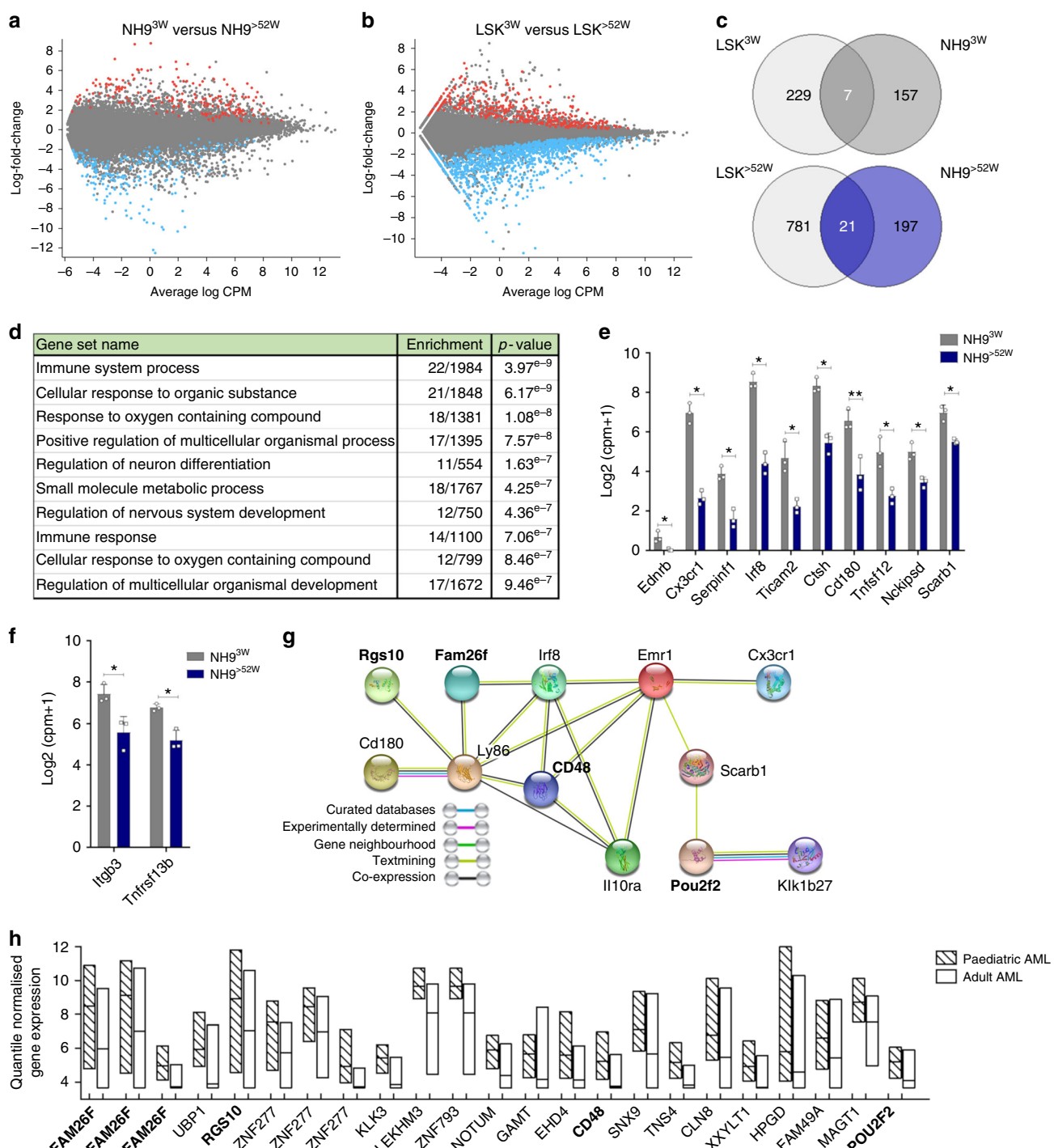

**Fig. 5** Differential enrichment of pathways in age-defined AML. MA plot highlighting significantly DEGs from pairwise comparisons between **a** myeloid leukaemia cells from NH9[3W] and NH9[>52W] and **b** healthy LSKs of 3w (LSK[3W]) and >52w (LSK[>52W]) origin. 1 RNA sequencing experiment with $n = 3$ biological replicates per group. DEGs with adjusted $p$-value <0.05 and LFC ≥1.5 highlighted in red (up in NH9[3W]) and in blue (up in NH9[>52W]). **c** Venn diagram showing overlapping genes from LSK[3W] and NH9[3W] DEGs (top) and LSK[>52W] and NH9[>52W] DEGs (bottom). **d** Top ten enriched GO pathways using DEGs upregulated in NH9[3W] compared to NH9[>52W] samples but not in LSK[3W] (157 DEGs). Analysis performed on MSigDB using C5 module (GO Biological Processes) with an FDR $q$-value of <0.05. **e** DEGs upregulated in NH9[3W] compared to NH9[>52W] but not in LSK[3W], in common to 5 or more of the top 10 pathways identified in **d**. **f** Graph of genes from cluster (ii) (See Supplementary Figure 3c, d) that are significantly upregulated in NH9[3W] compared to NH9[>52W]. **g** Protein interaction network of selected proteins whose gene expression was significantly upregulated in NH9[3W] compared to NH9[>52W] but not in LSK[3W], predicted by STRING. Different types of associations between the selected proteins are represented by different line colours as indicated in legend. Genes highlighted in bold were also identified as significantly enriched in the human paediatric dataset in **h**. **h** Quantile normalized gene expression of NH9[3W] DEGs that were significantly enriched in human paediatric AML samples ($n = 237$) compared to human adult AML samples ($n = 537$). Floating bars represent min and max values with mean. Individual probes shown, some genes with multiple probes present. Genes highlighted in bold were also identified in STRING protein hub **g**

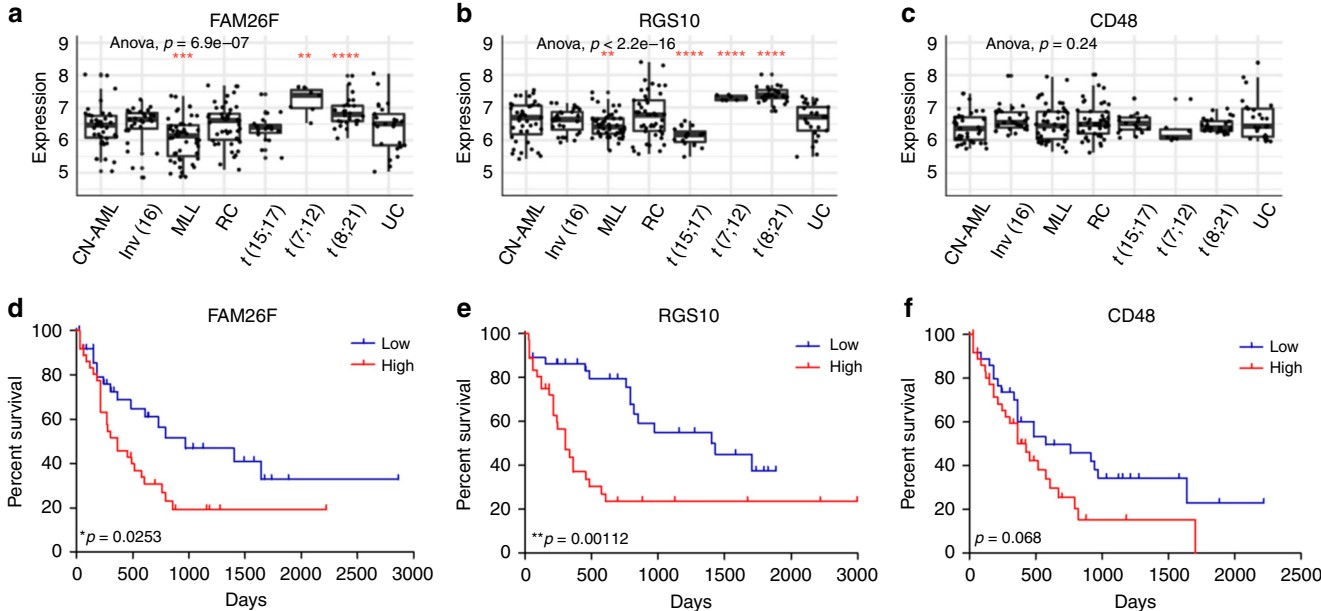

**Fig. 6** Genes in paediatric signature associate with AML subtypes and prognosis. Expression of **a** FAM26F, **b** RGS10 and **c** CD48 in paediatric AML patients (GSE17855) categorised by AML subtypes—cytogenetically normal AML (CN-AML, $n = 39$), AML with inv[16] ($n = 27$), MLL rearranged AML (MLL, $n = 47$), AML with remaining cytogenetics including $t(6;9)$, monosomy 7, trisomy 8 and complex karyotype (RC, $n = 45$), AML with translocations $t(15;17)$ ($n = 19$), $t(7;12)$ ($n = 7$) and $t(8;21)$ ($n = 28$) and patients with unknown cytogenetics (UC, $n = 25$). Significance determined by one-way ANOVA and one sample $t$-test using the ggpubr package (https://CRAN.R-project.org/package = ggpubr) *$p < 0.05$, **$p < 0.01$, ***$p < 0.001$, ****$p < 0.0001$. **d–f** Kaplan–Meier survival analysis of overall survival of AML patients ($n = 150$) in the TCGA L-AML dataset with high (top quartile, $n = 37$) and low (bottom quartile, $n = 37$) expression of **d** FAM26F, **e** RGS10 and **f** CD48. Significance determined by overall log rank *$p < 0.05$, **$p < 0.01$

paediatric AML gene signature included genes involved in immune cell interaction FAM26F and CD48, as well as a G-protein signalling regulator RGS10.

We next examined the expression levels of our human paediatric gene signature within the different cytogenetic subtypes of AML from the paediatric dataset (GSE17855, $n = 237$). Both FAM26F and RGS10, whose expression correlated with poor clinical outcome, were found significantly enriched in AML with $t(7;12)$ (MNX1/ETV6) and $t(8;21)$ (RUNX1/RUNX1T1) (Fig. 6a, b) whereas expression of CD48 did not correlate with any AML subtype (Fig. 6c). Of the remaining genes, the expression pattern of HPGD was strikingly similar to that of RGS10 being significantly higher in AML with $t(7;12)$ and $t(8;21)$ (Supplementary Figure 5). To investigate the prognostic significance of the genes in our human paediatric AML signature we used the OncoLnc platform to mine the available AML dataset with prognostic information, the TCGA L-AML dataset of 150 adult AML patients. This analysis shows patients with high expression of FAM26F or RGS10 have significantly poorer overall survival (Fig. 6d, e). High CD48 expression is also correlated with poor prognosis, although this did not reach our significance threshold (Fig. 6f). Patients with high expression of XXYLT1 had significantly better overall survival than those with low expression, whereas expression levels of the remaining genes did not significantly impact patient prognosis (Supplementary Figure 4).

Our findings support the hypothesis that AMLs of paediatric origin have a distinct molecular profile impacting on innate immune cell pathways, indicating differential interactions with BM niche components compared to adult AML. Gene expression profiling in AML is a well-established way to provide valuable diagnostic insight into predicting clinical outcome and our data show that the expression of FAM26F and RGS10, which are

significantly enriched in two AML subtypes, correlates with significantly poorer clinical outcome.

## Discussion

Current therapy for paediatric AML has been extrapolated from adult AML treatment, which assumes that the disease in children and adults are similar. The recent TARGET AML dataset showed distinct features of age-specific AML disease, calling for increased understanding to drive paediatric-specific therapy[12]. This study provides a murine model that can recapitulate paediatric disease features, a model that can be used to drive forward with new modes of paediatric-specific therapies. We used an age-defined in vivo leukaemia model to address the hypothesis that the leukaemia cell age influences disease biology and aggressiveness. Our results have shown that significant biological differences exist in AML derived from a young or adult cell of origin. Transplantation of NH9$^+$ adult (10w and >52w) LSKs resulted in AML in 100% of recipient mice. In contrast, transplantation of NH9$^+$ young (FL and 3w) LSKs resulted in a less penetrant disease with a significantly longer latency. Whereas the self-renewal ability of HSCs decreases with age, our model uses a bulk LSK population that includes multipotent and more restricted progenitors with varying self-renewal[33]. It is not known how the self-renewal capacity of these cell populations changes with age. Indeed, we observed increased homing to the BM and spleen with the older NH9-transduced cells. Despite initial homing differences, however, similar levels of GFP (donor cell burden) were observed in the PB after 16 weeks. It would be interesting to see whether the age of the recipient mouse (in our case it was 6–8 wks) had an impact on the levels of homing, engraftment and disease generated from the different age donor cells. In addition to AML, NH9

drove ALL and MPAL diseases from young LSKs but not adult. These findings are supported by observations in human leukaemia where lineage promiscuity has been observed more commonly in paediatric leukaemia[34,35]. Thus far the NH9 fusion oncogene has been exclusively linked with myeloid leukaemias, but other Nup98 fusion proteins have been identified in cases of human paediatric T-ALL[36–38]. NUP98-HOXD13 can induce both AML (60%) and T- and B-ALL in vivo[39,40]. The multi-lineage disease output that we observe from a young cell of origin may be accounted for by strong intrinsic lymphoid programmes inherent in younger LSKs that are only revealed in the BM microenvironment. Together our data show that the age-related transcriptional landscape of cells at the time of transformation has a significant impact on the biology of the resultant disease.

Contrary to our in vivo results, our in vitro data showed no difference in the lineage or transformation potential between the different ages of NH9+ LSKs. The disparity between these models suggested that the BM microenvironment plays a role in the age-related differences in leukaemia observed. Upon transplantation into secondary recipients, we observed remodeling of the BM microenvironment by adult AML blasts (NH9>52W) with a significant increase in marrow ECs, and OBs and MSCs in the compact bone. This correlates with several recent publications that have identified remodelling of the BM stroma by leukaemic blasts[19,41]. Leukaemic cells are protected and supported by interacting with the BM niche, and disease relapse is caused by the persistence of leukaemia cells that evade treatment. In humans, adult AML patients have higher relapse rates than children with AML. The ability of adult leukaemic cells to remodel the BM stroma in our model suggests that specifically targeting interactions between AML cells and the niche in adults may improve therapy outcome and reduce the rate of relapse. In contrast, AML blasts from NH9[3W] mice had no effect on the stroma of recipient mice upon transplantation. The BM niche is a complex microenvironment consisting of multiple cell types beyond stromal ECs, MSCs and OBS, including immune cells, adipocytes and neural cells. Our data indicate that NH9[3W] blasts may interact with non-stromal elements of the niche, and elucidating these interactions could identify druggable targets specific to paediatric AML.

Hierarchical clustering analysis of genes encoding receptors involved in cancer-stroma interactions identified enrichment of unique gene clusters associated with either NH9[3W] or NH9[>52W] cells, confirming our in vivo identification of differential interaction between these age-defined leukaemic blasts and the stroma. GO pathway analysis of DEGs revealed significant enrichment of metabolic pathways in NH9[>52W] cells compared to healthy LSK[>52w], while in the young myeloid blasts (NH9[3W]) we observed significant enrichment of immune system pathways, and upregulation of genes involved in TLR signalling compared to both healthy LSK[3w] and NH9[52w]. Analysis of DEGs upregulated in NH9[3W] AML blasts using STRING revealed a hub of innate immune cell interactions surrounding CD180. CD180 is a TLR-like receptor expressed on B cells and monocytes, but not T cells, where it forms a complex with MD-1 (Ly86) to regulate TLR2/4 LPS responses[42]. MD-1 was also significantly upregulated in NH9[3W] myeloid blasts. Activation through CD180 mediates pro-survival and proliferation signals in both normal and leukaemic B cells in vitro[43]. CD180 has been identified as a marker of LSCs in AML[44] and along with CCR2, it was also recently identified as a whole blood biomarker for BRD4 inhibitors[45].

Comparison of paediatric and adult AML datasets revealed a human paediatric gene signature with 19 of the 164 murine DEGs (11.6%) significantly enriched in paediatric AML samples, which included several genes involved with immune cell interaction. Regardless of age, AML patients with high expression of 2 genes

in our paediatric signature—FAM26F and RGS10—had significantly poorer overall survival rates. RGS10 is expressed in the brain, lymph nodes, testis and bone marrow[46] where it has been shown to play a key role in regulating inflammation. While little is known about the role of RGS10 in AML, in a meta-analysis of over 2700 AML patients across 25 published datasets RGS10 was identified in the top 25 most common differentially regulated genes—along with well-established AML associated genes such as HOXA9, MEIS1 and RUNX1T1[47]. FAM26F is a cation channel that regulates immune response by trafficking substances into or between cells[48]. It has recently been proposed as an early diagnostic marker as its increased expression is indicative of IFN-γ mediated immune responses[49]. While AML with t(7;12) is rare in children as a whole (~3%) it is most commonly seen in infant (0–1 years) AML with one-third bearing the translocation[50]. It is a cryptic abnormality which can't be diagnosed using traditional karyptyping methods and is associated with very poor clinical outcome. While t(8;21) is in fact considered a good risk AML, not all patients in this subgroup do well and these genes may indeed determine whether someone with t(8;21) does respond well or not to treatment. Here we identify 2 genes, RGS10 and FAM26F which are significantly enriched in this subset of paediatric AML patients which could act as a biomarker for diagnoses.

The lifetime risk of developing cancer is determined by a combination of inherited predispositions, environmental factors, and random mutations arising during DNA replication. While cancer is widely considered a disease of the aged, it is the leading cause of death in children and young adults. The appropriateness of the current practice for some cancers of extrapolating treatments across the age spectrum is dependent on the assumption that the same aetiology underlies cancer in the young and old. Our results show a specific transcriptional landscape of paediatric AML that significantly distinguishes paediatric disease biology and is supported by murine modelling. We identify FAM26F and RGS10 as significantly enriched in paediatric AML associated with the AML subtypes t(7;12) and t(8;21) and these genes are linked with poor clinical outcome. These are potentially useful biomarkers in paediatric AML and their function and upregulation in paediatric AML of much interest. These findings underpin the necessity to advance the development of age-appropriate and biologically driven precision medicine for the eradication and cure of AML in children.

## Methods

**Retroviral production.** Plasmids were constructed in the MigR1 backbone, for the production of retrovirus. The NUP98HOXA9-MSCV-IRES-GFP and FLT3-ITD (N51)-MSCV-IRES-GFP vectors were a gift from Brian Huntly (Cambridge Institute for Medical Research, University of Cambridge, UK). The AML1ETO-MSCV-IRES-GFP vector was a gift from James Mulloy (Cincinnati Children's Hospital Medical Center, University of Cincinnati College of Medicine, USA). Retrovirus was produced by calcium phosphate transfection of constructs with packaging vectors pCGP and pHIT123 in HEK293T (ATCC) cell line. Retrovirus containing supernatants were harvested at 24, 36 and 48 h post transfection. Viral titres were measured by the transduction of 3T3 (ATCC) cells in the presence of Polybrene 4 μg mL−1 (Sigma-Aldrich) and percentage of GFP determined by flow cytometry.

**Retroviral transduction of primary cells.** Murine BM and FL cells were retrovirally transduced with titre-matched retrovirus to express GFP tagged oncogenes. BM and FL cells were labelled with ckit (CD117) microbeads (Miltenyi Biotec) to enrich cKit cells by passing through a MS Columns (Miltenyi Biotec) in the magnetic field of a MACS Separator (Miltenyi Biotec). cKit enriched BM or FL cells were pre-stimulated overnight in pre-stimulation cocktail (DMEM 15%FBS, 1 mg ml−1 Penicillin/Streptomycin (PS) and 2 mM L-Glu) supplemented with 10 ng ml−1 recombinant murine (rm) IL3, 10 ng ml−1 rmIL6 and 100 ng ml−1 rmSCF (Peprotech). For transduction, cells were resuspended in activation cocktail (DMEM 15%FBS, 1 mg ml−1 PS, 2 mM L-Glu, 10 ng ml−1 rmIL3, 10 ng ml−1 rmIL6 and 10 ng ml−1 rmSCF, and 4 μg ml−1 Polybrene) and virus was quickly thawed and added. The final cell concentration was 0.5–2 × 10⁶ cells/mL. Two 90 min spinoculations were performed 6–12 h apart. After 48 h transduced cKit

enriched BM or FL cells were FACs sorted for LSK, CMP and GMP populations (see Supplementary Figure 7).

**Serial colony forming cell (CFC) Assays**. FACs sorted, transduced LSK, CMP and GMP populations were resuspended in Methocult M3434 (Stem Cell Technologies). After 7–10 days, colonies were counted, categorised and images taken on an EVOS XL Core Imaging System (Fisher Scientific). Colonies were categorised based on their morphology as per StemCell Technology Methods and were of 4 distinct phenotypes: granulocyte erythroid macrophage megakaryocyte (GEMM), granulocyte macrophage (GM), granulocyte (G) and macrophage (M) (Supplementary Figure 1A). Cells were replated three times (P1–3).

**OP9 co-culture assay**. Transduced FL HSPCs (LSK, CMP and GMP) were plated onto a sub-confluent layer OP9 cells (MTA from Juan Carlos Zuniga-Pflucker) in OP9 media supplemented with 10 ng $\mu l^{-1}$ IL3, 10 ng $\mu l^{-1}$ IL6, and 100 ng $\mu l^{-1}$ SCF. Media were refreshed every 2 days and cells were replated to a fresh OP9 layer every 4 days. At each replating, FL cells were counted and GFP expression determined. If FL cells had expanded to a density of $\geq 1 \times 10^6$ cells per mL a 1:10 dilution was carried out. To analyze myeloid and lymphoid potential, transduced NH9 LSK cells at P2 from FL, 3w, 10w and>52w were plated onto a sub-confluent layer OP9/ OP9-DL1 cells in OP9 media supplemented with 5 ng ml$^{-1}$ Flt3 and 1 ng ml$^{-1}$ IL7. Media were changed at 2 days and cells were analyzed by flow cytometry at 4 days.

**BMT experiments**. Donor LSKs transduced with GFP-tagged NH9 retrovirus and expanded in methylcellulose were collected at P2. $1 \times 10^6$ GFP$^+$ cells were intravenously transplanted into sub-lethally irradiated 6–8w old recipient mice (450rads). For BM reconstitution experiments animals were monitored for engraftment by GFP% in PB and clinical signs of leukaemia (piloerection, tachypnoenea, reduced mobility, weight loss, PB total white cell count >20 × 10$^6$). PB parameters were analysed on a HemaVet 950 (Drew Scientific). Animals were killed on onset of leukaemia or 12 months post BMT. For secondary transplantation, BM blasts from primary recipients that developed AML were intravenously transplanted into sub-lethally irradiated recipient mice (B6.SJL). For homing experiments, $2 \times 10^6$ GFP NH9 transduced donor cells were labelled with CellTrace$^{TM}$ Violet (CTV) (Invitrogen) and intravenously transplanted into sub-lethally irradiated 6–8w old recipient mice (550rads). Mice were killed 18 h post transplant, and BM, PB and spleen assessed for the presence of GFP + CTV+ donor cells by flow cytometry. Percentage of cells homed was obtained from the absolute number of donor cells, calculated from the percentage of engraftment and total cell number per organ, and the number of the cells injected.

**Flow cytometry analysis and sorting**. Peripheral blood was collected into an EDTA lined Microvette® capillary blood tube (Sarstedt). Bones were crushed in PBS with 2% FBS and the cells were filtered through a 40 μm strainer to isolate the central marrow. To isolate cells in the endosteal niche, the compact bone fragments were digested with 1 mg ml$^{-1}$ collagenase type IV (Gibco) and 2 mg ml$^{-1}$ Dispase II (Sigma) at 37 °C, twice for 15 min with agitation. Splenic and thymic cells were filtered through a 40 μm strainer. In all tissues red blood cells (RBCs) were lysed in RBC lysis buffer (155 mM NH$_4$Cl, 10 mM KHCO$_3$, 0.1 mM EDTA). Single cell suspensions were incubated with antibody cocktail at 4 °C for 20 min.

For myeloid staining samples were stained with CD11b PE (eBioscience 12-0112-83) or CD11b APC (eBioscience 17-0112-83), Gr1 PerCP-Cy5.5 (eBioscience 45-5931-80) or Gr1 eFlour 450 (eBioscience 48-5931-82), F4/80 APC (eBioscience 17-4801-82), cKit ACP-Cy7 (Biolegend 105825), and CD34 biotin (eBioscience 13-0341-82) and Streptavidin-PE-Cy7 (eBioscience 25-4317-82) at dilution 1:200.

For lymphoid staining samples were stained with CD4 APC (eBioscience 17-0042-83), CD8a PE (eBioscience 12-0081-82), B220 PerCP-Cy5.5 (eBioscience 45-0452-82) or B220 eFluor 450 (eBioscience 48-0452-82), CD19 APC-Fluor780 (eBioscience 47-0193-80) or CD19 APC (eBioscience 17-0193-82) and CD3e eFluor 450 (eBioscience 48-0031-82) at dilution 1:200.

For progenitor cell isolation GFP-tagged cKit enriched BM or FL was stained with cKit APC (eBioscience 17-1171-82), Sca1 PE-Cy7 (eBioscience 25-5981-82), CD34 biotin and Streptavidin-PE (eBioscience 12-4317-87), CD16/32 APC-Cy7 (BD 560541) and lineage markers: eFlur®450 conjugated CD3, CD4 (eBioscience 48-0042-82), CD8a (eBioscience 48-0081-82), CD11b (eBioscience 48-0112-82) (omitted for FL and 3w), GR1, B220 (eBioscience 48-0452-82) and Ter119 (eBioscience 48-5921-82).

For isolation of LSK cells (transplanted in control mice and for RNA-seq analysis), BM cells were stained with lineage cocktail with the following biotinylated antibodies: CD3 (eBioscience 13-0032-82), CD4 (eBioscience 13-0041-85), CD8a (eBioscience 13-0081-85), B220 (eBioscience 13-0452-85), CD11b (eBioscience 13-0112-85), Gr-1 (eBioscience 13-5931-85) and Ter119 (eBioscience 13-5921-82), followed by anti-biotin MicroBeads staining for lineage cells depletion in LS Columns (Miltenyi Biotec) in the magnetic field of a MACS Separator (Miltenyi Biotec). The lineage-negative population were stained with Sca1 PE-Cy7, cKit ACP-Cy7 and Streptavidin V450 (BD 560797) at dilution 1:200.

For identification BM stromal cells were stained with CD45 bio (eBioscience 13-0451-82), Streptavidin PerCP-Cy5.5 (eBioscience 45-4317-82), Ter119 eFluor 450,

CD31APC (eBioscience 17-0311-80), CD51PE (eBioscience 12-0512-82), PDGRa PE-Cy7 (eBioscience 25-1401-80) at dilution 1:200.

Where applicable dead cells were also excluded using either a 4',6-diamidino-2-phenylindole (DAPI) at dilution 1:1000 (Sigma-Aldrich D9542). Flow cytometry data were acquired using a FACSCanto II (BD Biosciences) and cells were sorted using a BD FACSAria III (BD Biosciences). Flow cytometry data were analysed using FlowJo software v7 (Treestar Inc).

**Histopathology**. Sternal sections were decalcified with 5.5% EDTA solution containing 10% formaldehyde for 7–8 days. Decalcified sterna were embedded in paraffin and sections cut at 2.5 μm and fixed onto microscopy slides. Immunohistochemistry (IHC) was performed with anti-MPO antibody (Dako Agilent A0398) at dilution 1:3000 or anti-CD3 (Dako Agilent A0452) antibody at dilution 1:100. Tissue sections were dewaxed in xylene and descending alcohols and rehydrated in distilled water; endogenous peroxidase was quenched by immersion in 3% hydrogen peroxide (H$_2$O$_2$) in PBS for 5 min.

**Transcriptional analyses**. RNA was isolated using either the RNeasy Mini kit (Qiagen) or Arcturus PicoPure kit (Fisher Scientific) and total RNA converted to cDNA using the High Capacity cDNA Reverse Transcription Kit (Invitrogen) as per manufacturers' instructions. Targeted gene expression was performed on a 48.48 dynamic array (Fluidigm) in accordance with manufactures' protocol and gene expression analyzed using standard techniques. For RNA sequencing, RNA libraries were prepared by polyA selection using the TruSeq stranded mRNA kit (Illumina). Paired end RNA-seq was performed on a NextSeq$^{TM}$ 500 platform (Illumina) with a read and fragment length of 75 bp and 150 bp, respectively, to a depth of 33 million reads. Reads were aligned using the standard Stemformatics RNA-Seq data processing pipeline (https://www.stemformatics.org/Stemformatics_data_methods.pdf). Differential gene expression was performed using limma, voom and Glimma packages in R (https://f1000research.com/articles/5-1408/v2). Significance determined by adjusting for false discovery rate using Benjamini-Hochberg method as used by limma. List of DEGs are provided in supplementary data (Supplementary Data 1-2). Pathway analysis was performed using EGSEA package in R (https://f1000research.com/articles/6-2010/v1). Raw data are available at GEO with ID GSE107662 (AML) and GSE122066 (Normal LSKs), and can be accessed on the Stemformatics platform at https://www.stemformatics.org/datasets/search?ds_id=7127# and https://www.stemformatics.org/datasets/search?ds_id=7311#.

Gene lists of DEGs were interrogated for GO pathway enrichment using the Molecular Signatures Database (MSigDB) (Broad Institute)[51]. Overlaps were computed with a false discovery rate (FDR) $q$-value below 0.05 using the C5 GO module. Where indicated, analysis was carried out using Stemformatics data analysis platform[30]. When analyzing GSE17855 and GSE6891 standard batch correcting methods sva[52] and ruv[53] over-or-under corrected the data. Instead an alternative approach was used. A set of reference genes common to both datasets, with low variance and mean within each datasets were identified and used for normalization. First the datasets were quantile normalised so that their value ranges were compatible, and the probes with low variance and mean were identified to be used as the reference genes. The 164 DEGs upregulated in NH9$^{3W}$ compared to NH9$^{>52W}$ were then mapped to their human orthologues and corresponding probe identifiers in each dataset. Comparing the difference between each probe and the reference probe identified genes significantly enriched in the human paediatric AML dataset (where the $p$-value is the result of the $t$-test for the null hypothesis where the means of these differences are equal).

Prognostic significance was determined using the OncoLnc platform (http://www.oncolnc.org/) which links to TCGA survival data[54]. Selected genes were analysed within the L-AML dataset of 150 AML patients. Patients were filtered for high (top quartile, $n = 37$) or low (bottom quartile, $n = 37$) expression of each gene. AML subtype graphs were generated using ggplot2 package in R (http://www.R-project.org/). For multiple groups comparison one-way ANOVA and $t$-test were calculated using the ggpubr package in R and each of the genes' mean expression was compared to the base-mean of the rest.

**RT-PCR analysis**. RNA was extracted from P1 and P3 NH9 and P1 MigR1 cells from different ages using the RNeasy® Mini kit (Qiagen). Reverse Transcriptase PCR was performed following the protocol from High Capacity cDNA Reverse Transcription Kit. Primer names and sequences provided in Supplementary Table 1 and uncropped gels provided in Supplementary Fig. 6, in Supplementary Information file.

**Quantification and statistical analysis**. GraphPad Prism v7 (GraphPad Software Inc.) was used for statistical analysis and production of graphs. When 2 groups were compared and $F$-test was performed to determine sample variance, then an unpaired, two-tailed Student's $t$-test was used to determine significance. When more than 2 groups with 1 variable were compared, a one-way ANOVA with Bonferroni post-test for FDR was used to determine significance. When more than 2 groups with two variables were compared, a two-way ANOVA with Bonferroni post-test for FDR was used to determine significance. For comparison of survival

curves the Log-rank test was used. Significance was attained with a p-value (or adjusted p-value for FDR) of <0.5 (*), <0.01 (**), <0.001 (***) and <0.0001 (****).

**Animals and ethical compliance**. C57BL/6 (CD45.2+, B6) mice were purchased from Charles River UK. B6.SJL-Ptprca/BoyAiTac (CD45.1+, B6.SJL) mice bred and housed in the Beatson Biological Service and Research Units. All animal experiments have been carried out at the University of Glasgow and were approved by United Kingdom (UK) Animal Ethical Committees and performed according to UK Home Office project license 60/4512 (Animal Scientific Procedures Act 1986) guidelines.

## Data availability

There are no restrictions on data availability. The RNA-Seq data containing NH9 samples are available at Stemformatics dataset [7127]. The raw data have been deposited in NCBI's Gene Expression Omnibus (GEO), with accession number [GSE107662]. The RNA-Seq data containing wild-type LSK samples are available at Stemformatics (dataset [7311]), and the raw data have been deposited in GEO with accession number [GSE122066]. These raw data are associated with Figs. 4a, b, 5a–h Supplementary figures 2a–e, and 3a–g.

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

## Acknowledgements

We acknowledge the technical assistance of Jennifer Cassels and the flow cytometry core funding from the Kay Kendell Leukaemia Fund and The Howat Foundation. We thank Karen Dunn for technical assistance with animal experiments and the Cancer Research UK Glasgow Centre (C596/A18076) and the BSU facilities at the Cancer Research UK Beatson Institute (C596/A17196). We thank Francesco Marchesi, Lynn Stevenson, Lynn Oxford and Frazer Bell (Veterinary Diagnostic Services, School of Veterinary Medicine, University of Glasgow) for their technical assistance with histology and immunohistochemistry. We thank Tyrone Chen for processing all relevant datasets. We thank all staff at the Paul O'Gorman Leukaemia Research Centre for helpful suggestions and advice. We thank support from Children with Cancer UK (to K.K.), the Howat foundation (to K.K., C.O'C., A.C.), Yorkhill Children Cancer, Yorkhill Children's Leukaemia Research Fund (to E.S., S.C., A.P.) and the Friends of Paul O' Gorman (to J.B.S.).

## Author contributions

S.C., C.O'C., A.C., E.S., A.P. and PJ. performed experiments, analysed data, and made the figures. C.O'C., J.C., J.B.S. and C.W. facilitated and performed computational analysis. C. W. and B.G. supervised the work. K.K. designed the experiments, interpreted the data and supervised the work. C.O'C., A.C. and K.K. wrote the paper.

## Additional information

**Competing interests:** The authors declare no competing interests.

