## [Peer Review File · Nature Communications]

Reviewers' comments:

Reviewer #1 (Remarks to the Author):

The age-specific transcriptional landscape of acute myeloid leukaemia reveals distinct paediatric disease biology

Shahzya Chaudhury, Caitríona O'Connor, Ana Cañete, Áine Prendergast, Jarny Choi, Pamela Johnston, Christine A. Wells, Brenda Gibson, Karen Keeshan

This is a remarkably comprehensive study demonstrating that the age of the hematopoietic stem/progenitors where genomic alteration evolves determines latency of leukemic evolution as well as the nature of the leukemia that develops. The authors use murine hematopoietic cells obtained from

Investigators isolated murine lin-sca1+cKit+ (LSK) cells from prenatal and postnatal mice (different ages) to represent infant/prenatal origins (fetal liver (FL)), childhood (BM from 3 week(w)), young adult (10w) and middle/old age (>52w) C57BL/6 mice. These cells of different age were transduced with NUP98/HoxA9 and the resultant cells were studied in vitro and in vivo for leukemic transformation, latency of transformation and type of the resultant leukemia. It is remarkable that the authors demonstrated vast differences in type of leukemia based on the age of the cells that were transformed, where the transformed older cells led to myeloid leukemia transformation, whereas fetal cells showed plasticity and developed AML, ALL or MPAL. They also demonstrate that transduction of older cells is associated with shorter latency and more rapid leukemic progression. Additional transcriptional analysis demonstrated different enriched pathways with younger cell of origin enriched in development or immune signature, whereas older cells associated with cell-cell interaction. They extended their studies into cell-microenvironment interactions and again demonstrating differential expression of genes in the different age mice.

This very comprehensive and highly significant finding illuminates the significance of the cell of origin in malignant transformation and may indeed shed some light into the significant diversity of AML and the vast differences observed in infant AML vs. other childhood AML vs. adult/elderly AML.

Minor comments to be added to the manuscript:

1. It is important to mark a clear distinction between myelo/lympho-proliferative disorder and true leukemic phenotype. Additional language/figs to clearly demonstrate that the observed phenotype is true leukemia vs. –proliferative disorder would be helpful.
2. Clear molecular evidence of presence of NUP98/HoxA9 transcript in the transduced cells would help determine whether the observed phenotype is due to the fusion vs. disruption of the genome/transcript.
3. Transcriptional profiling of fetal vs. older cells (prior to transduction) may shed light into the observed differences into the baseline differences between the LSK cells in fetal vs. older mice.
4. Addition of patient data comparing the transcriptome signature in infant AML vs. older would significantly enhance the manuscript. The fact that malignant transformation in fetal cells is associated with immune signature is intriguing.

Reviewer #2 (Remarks to the Author):

General comments-summary:

The authors use a well-established mouse model to characterize age-related differences between pediatric and adult acute myeloid leukemia (AML). As a model system they transplant hematopoietic stem and progenitor cells (LSK) with different age (fetal liver, 3, 10 and >52weeks) that retrovirally express a rare fusion gene (NUP98-HOXA9, NHA9) resulting from a t(7;11) translocation that occurs in pediatric and adult AML into irradiated mice. They found similar in vitro transformation potential of LSK of different age (Fig.1). However, they observed that mice

transplanted with young NHA9 expressing LSK developed the disease later than those obtaining old cells (Fig.2). Adult NHA9-expressing LSK induced disease was purely myeloid (AML) whereas disease induced by young NHA9+ LSK presented as AML, ALL or a mixed phenotype (MPAL) differentially expressing lineage associated genes (Fig.3). Differentially expressed genes (DEG) between AML induced by NHA9-expressing LSKs were determined. Most interestingly, age-origin-related changes in the bone marrow microenvironment were observed. (Fig.4). They interrogated DEG of NHA9 AML from young and old LSK with different bioinformatics data platforms to deduce common genes. In addition they particularly looked for aberrantly expressed ligand-receptor interactions. Finally they compared DEG from young NHA9 AMLs with profiles from primary AMLs and identified 20 genes that seemed higher expressed in pediatric than adult patients (Fig.5). The study addresses a very important topic that is currently not well understood. The study is original and novel. Overall the MS is well written, clear and easy to follow. Figures are carefully prepared and mostly well organized. There are some points that need clarification; in addition some additional experiments could clearly improve the quality of this work.

Detailed comments

Major points:

1. The finding that the age of the transplanted LSKs retrovirally expressing the NHA9 fusion determines differential changes in the microenvironment of transplanted mice (associated with a different phenotype) is interesting. The authors describe them only in the setting of "secondary" transplants. Why, were they not detected in the primary transplants? Or was the phenotype different (AML, ALL, MPAL) in the primary transplant and more homogenous (AML) in the secondary? It is not clear from the text. In such a case, one could only compare similar phenotypes, as disorders affecting different lineages most likely also induce different changes in the microenvironment. Independent from that, the situation seems not to be so clear as presented: in Fig. 4, the authors claim that there were no statistically different levels of GFP expression (correlating with the fusion) in transplanted young (3w) and old (52w) LSK (Fig.4D). However, although the number of mice was rather low (maybe too low for the observed heterogeneity), one can clearly see two groups of mice transplanted with old LSKs: 4 with about 80% and 4 with about 20% of GFP-expressing cells (which will most likely be statistically different when compared to each other). This seems important, as similar splitting in high and low is seen in the analysis of the central marrow (Fig.4F), and the endosteal niche (Fig.4H). Would this observation suggest that expression levels of the fusion (GFP) is somehow linked with differential alteration of the microenvironment? The authors have to clarify these points.
2. The authors try to deduce mechanistic insights of the microenvironmental dynamics in leukemias induced by transplantation of NHA9 expressing LSKs of different age by interrogation the blast-derived DEG signatures with different bioinformatics databases. However, it is not clear why they not directly compare DEG in stroma cells (MSC, Endo, OBs) in diseased mice with the DEG expressed by the leukemic cells? In the current situation, all findings are indirectly deduced and also not validated in the primary system.
3. The fact that transplantation of methylcellulose (passage 2, P2)-expanded LSK expressing NHA9 cells induces phenotypically different leukemia after different latencies is striking. The authors suggest that independent of the origin, cells express similar levels of the fusion (deduced by MFI-GFP, Fig.1B) and similar lineage-related surface markers at P3 (Fig.1H) (the latter might not have reached significance simply due to the relatively large standard deviations). Additional controls would help to better understand what is going on: in particular one would like to see what was injected? Were there any age-related differences already detectable in the P2-derived cells before injection? The authors could compare expression of the NHA9 fusion by Q/PCR (or immunoblotting), immunophenotype (including lymphoid markers), and the lineage-specific clonogenic output of the cells in MC. Based on their observations in vivo, "young" NHA9 expressing LSK should keep the potential to produce lymphoid cells whereas the "old" NHA9 expressing LSK would be clearly be biased to the myeloid lineage. In addition, did the authors also compare

homing and grafting of NHA9-expressing LSKs (P2) of different age? Furthermore, did the authors also consider that some of the observed differences might be the consequence of target cells bias of transduction by the MSCV virus related to developmental age of the cells? Should this possibility not be discussed?

Minor points

1. Data shown in Fig. 2D should be shown as dot-blot not as bar plot, simply as the data points of the mice receiving "young" cells seemed much more heterogeneous (error bar) as those that received "adult" cells (no error bar?).

1. Fig.5G compares DEG of NHA9-3w AML with human AML (pediatric/adult): 2 out of 20 genes (CD48, FAM26F) are discussed. But what is the value of this analysis if the clinical and/or functional consequences remain completely unclear? Would it not be better to select a few and a) show significance value, b) look for clinical/biological correlations or functionally validate them in the primary experimental system?

2. Fig.4B shows a PCA analysis of DEG of leukemia that developed after BMT of NHA9-expressing LSK. Why are the NHA9>52w closer to NHA9FL than to NHA910w? How did the immunophenotypes (AML vs. ALL vs. MPAL) influence the DEG profiles?

3. Figure 3D, Immunohistochemistry: with this size, the details are not visible: the authors could insert part of the picture at higher magnification?

4. Is the title ideal? it is not only the transcriptional landscape but the biology of the obtained disease phenotype in general? I would reconsider the title.

4. Methods section: lines 419-421: Murine Stem Cell Virus, "MSCV" not "MSVC"?

Response to Reviewer #1

We thank the reviewer for their comments and for highlighting the comprehensive and significant nature of our work. Please find our response to their *minor* comments:

1. It is important to mark a clear distinction between myelo/lympho-proliferative disorder and true leukemic phenotype. Additional language/figs to clearly demonstrate that the observed phenotype is true leukemia vs. –proliferative disorder would be helpful.

Our response:

We have added text in the 3rd section of the results to clearly demonstrate this.

2. Clear molecular evidence of presence of NUP98/HoxA9 transcript in the transduced cells would help determine whether the observed phenotype is due to the fusion vs. disruption of the genome/transcript

Our response:

We have provided molecular evidence for the presence of NUP98/HoxA9. PCR using primers that cross the NH9 breakpoint was performed and presented in **Fig 1C and 1D**.

3. Transcriptional profiling of fetal vs. older cells (prior to transduction) may shed light into the observed differences into the baseline differences between the LSK cells in fetal vs. older mice.

Our response:

We have performed RNA-seq analysis on normal LSKs from 3w and >52W mice and all of **figure 5** analysis has been re-done omitting the genes that are differentially expressed between normal 3w and >52w LSKs. In addition, GO pathway analysis of normal 3w and >52w LSKs is shown as new supplemental **Fig S3D-E** showing that different pathways are enriched in normal young and old LSKs compared to the young and old leukaemias (**Fig 5D**).

4. Addition of patient data comparing the transcriptome signature in infant AML vs. older would significantly enhance the manuscript. The fact that malignant transformation in fetal cells is associated with immune signature is intriguing.

Our response:

Unfortunately, the samples in the Balgobind paediatric dataset (used in figure 5G of original manuscript) are not annotated with the patients age so we were not able to separate infant and older paediatric AMLs in this analysis. However to address the reviewers comment we have analysed the gene expression levels of our paediatric gene signature within AML subtypes of this dataset (new **Fig 5I-N, SFig4**). We have identified significant correlation between expression of FAM26F and RGS10 and a subset of paediatric AMLs with t(7;12) which is found in 1/3rd of all infant AMLs (<2yrs) while only 3% across all paediatric AML. These genes are also linked with poorer overall survival rates in AML (**Fig5 I-K**). We have provided additional text to reflect these new significant findings.

Response to Reviewer #2

We thank the reviewer for their comments and for highlighting that our study is original, novel and well presented. We have addressed the specific points raised by the reviewer which we agree has improved the clarity and quality of this work as follows:

1. The finding that the age of the transplanted LSKs retrovirally expressing the NHA9 fusion determines differential changes in the microenvironment of transplanted mice (associated with a different phenotype) is interesting. The authors describe them only in the setting of “secondary” transplants. Why, were they not detected in the primary transplants? Or was the phenotype different (AML, ALL, MPAL) in the primary transplant and more homogenous (AML) in the secondary? It is not clear from the text. In such a case, one could only compare similar phenotypes, as disorders

affecting different lineages most likely also induce different changes in the microenvironment.

Independent from that, the situation seems not to be so clear as presented: in Fig. 4, the authors claim that there were no statistically different levels of GFP expression (correlating with the fusion) in transplanted young (3w) and old (52w) LSK (Fig.4D). However, although the number of mice was rather low (maybe too low for the observed heterogeneity), one can clearly see two groups of mice transplanted with old LSKs: 4 with about 80% and 4 with about 20% of GFP-expressing cells (which will most likely be statistically different when compared to each other). This seems important, as similar splitting in high and low is seen in the analysis of the central marrow (Fig.4F), and the endosteal niche (Fig.4H). Would this observation suggest that expression levels of the fusion (GFP) is somehow linked with differential alteration of the microenvironment? The authors have to clarify these points.

Our response:

-We have clarified the first point in the text as it was only the myeloid leukaemias that were transplanted into secondaries therefore we have not linked AML/ALL/MPAL phenotypes to different microenvironment changes. We report the different microenvironment changes only with AMLs from different ages of origin.

-To clarify the point regarding GFP expression levels linked with changes, we have updated **Fig 4D-H** now using only mice with >30% GFP expression and marking each mouse with a different symbol so that they can be traced e.g., The blue diamond is the low NHA9^{52w}, and it does correspond in all the graphs as the low outlier. But if you track e.g., the blue square with cross (~60% GFP) and compare to the grey circle (~60%GFP) it is clear i.e. despite similar levels of GFP with leukaemias of the same phenotype (AML) there are different alterations to the microenvironment that are explained by the age of the cell of origin.

2. The authors try to deduce mechanistic insights of the microenvironmental dynamics in leukemias induced by transplantation of NHA9 expressing LSKs of different age by interrogation the blast-derived DEG signatures with different bioinformatics databases. However, it is not clear why they not directly compare DEG in stroma cells (MSC, Endo, OBs) in diseased mice with the DEG expressed by the leukemic cells? In the current situation, all findings are indirectly deduced and also not validated in the primary system.

Our response:

Yes it is true that we did not generate RNA-seq data from the different stromal cells (MSCs, Endo, OBs) from the mice transplanted with different age leukaemias. However, we have now performed RNA-seq analysis on normal LSKs from different ages and further deduced mechanistic insight of the age of leukaemia that is distinct from the healthy aged counterpart (new **Fig 5A-H**). A global interrogation into the DEGs of the MSCs, Endo, OBs from the different aged mice could potential omit cells that may be involved in the interaction/responsible for the DEGs. Therefore we have focused on DEGs that separate the young and old AMLs in both mice and humans (see new **Fig 5I-N**), which may alter the microenvironment through autocrine or paracrine mechanisms. Further transcriptional profiling of all the different stromal cells types would be an enormous undertaking and beyond the scope of this study.

3. The fact that transplantation of methylcellulose (passage 2, P2)-expanded LSK expressing NHA9 cells induces phenotypically different leukemia after different latencies is striking. The authors suggest that independent of the origin, cells express similar levels of the fusion (deduced by MFI-GFP, Fig.1B) and similar lineage-related surface markers at P3 (Fig.1H) (the latter might not have reached significance simply due to the relatively large standard deviations). Additional controls would help to better understand what is going on: in particular one would like to see what was injected? Were there any age-related differences already detectable in the P2-derived cells before injection? The authors could compare expression of the NHA9 fusion by Q/PCR (or immunoblotting), immunophenotype (including lymphoid

markers), and the lineage-specific clonogenic output of the cells in MC. Based on their observations *in vivo*, “young” NHA9 expressing LSK should keep the potential to produce lymphoid cells whereas the “old” NHA9 expressing LSK would be clearly be biased to the myeloid lineage. In addition, did the authors also compare homing and grafting of NHA9-expressing LSKs (P2) of different age? Furthermore, did the authors also consider that some of the observed differences might be the consequence of target cells bias of transduction by the MSCV virus related to developmental age of the cells? Should this possibility not be discussed?

Our response:

To address these points we have now adding the following data:

-To provide controls for the p2 cells that were injected that express similar levels of GFP, and to rule out any bias in transduction levels with NH9, we have provided molecular evidence for the presence of NUP98/HoxA9. As suggested by the reviewer, we have compared expression of NH9 fusion by QPCR using primers that cross the NH9 breakpoint and presented in new **Fig 1C and 1D** with no significant differences between young and old cells.

-We have included lymphoid markers in our immunophenotyping of p2 cells which are now included in a new supplemental figure (**Fig S1H**) showing that cells from all ages do not have significant expression or differences in lymphoid markers prior to transplantation.

-Additionally, we addressed the lineage-specific clonogenic output of the cells in MC by performing OP9 co-culture assay with cells from p2 (**Fig S1I**) showing the absence of lymphoid potential and the only significant difference being that fetal cells have more myeloid Gr-1 expression than other age groups (**Fig S1H-I**).

These data show that prior to transplantation we do not see any retainment of lymphoid potential, in agreement with our suggestion that the *in vivo* bone marrow microenvironment interactions and the transcriptional age of the cell explain the observed differences in disease *in vivo*.

-As suggested by the reviewer, we compared homing of NH9-expressing LSKs (P2) of different ages. These data show that the older cells are better at homing to the BM and spleen compared to the younger cells. Despite this, similar levels of GFP (no significant differences) were observed in the PB after 16 weeks. It would be interesting to see if the age of the recipient mouse had an impact on the levels of homing from the different age donor cells. This is beyond the scope however is provided as a discussion in the text.

Minor points

1. Data shown in Fig. 2D should be shown as dot-blot not as bar plot, simply as the data points of the mice receiving “young” cells seemed much more heterogeneous (error bar) as those that received “adult” cells (no error bar?).

Our response:

Figure 2D (now 2H?) has been changed to show the number of animals who developed disease for improved clarity.

2. Fig.5G compares DEG of NHA9-3w AML with human AML (pediatric/adult): 2 out of 20 genes (CD48, FAM26F) are discussed. But what is the value of this analysis if the clinical and/or functional consequences remain completely unclear? Would it not be better to select a few and a) show significance value, b) look for clinical/biological correlations or functionally validate them in the primary experimental system?

Our response:

Figure 5 has now been updated using the normal LSK RNA-seq data to omit genes that change with healthy age, and for the selected genes, we now show prognostic significance for the expression levels of these genes in AML (**Fig 5I-K**). Further, we have linked the expression with specific subtypes of paediatric AML (**Fig 5L-N**) providing clinical correlations as suggested by the reviewer. These data confirm that our selected genes are significant for paediatric AML. We provide additional data in a new supplemental **Fig S4 and S5** for the whole panel of genes in

our paediatric signature. This is provided in the results section and further discussed in the discussion section.

3. Fig.4B shows a PCA analysis of DEG of leukemia that developed after BMT of NHA9-expressing LSK. Why are the NHA9>52w closer to NHA9FL than to NHA910w? How did the immunophenotypes (AML vs. ALL vs. MPAL) influence the DEG profiles?

Our response:

The DEG profiles are from only the myeloid leukaemias so it is not influenced by lymphoid leukaemias. For the PCA analysis which is a global assessment and visual presentation of gene expression within samples and shows that there is some heterogeneity within the groups which is approximately the same as the intergroup heterogeneity. The DEGs between the age groups are a fraction of the total gene expression and thus the PCA does not reveal these distinctions. We have omitted the sentence stating that PCA revealed clustering as this was not technically accurate.

4. Figure 3D, Immunohistochemistry: with this size, the details are not visible: the authors could insert part of the picture at higher magnification?

Our response:

Figure 3D has been updated with x60 magnification images.

5. Is the title ideal? it is not only the transcriptional landscape but the biology of the obtained disease phenotype in general? I would reconsider the title.

Our response:

We have changed the title to reflect this comment. The title is now "Biological and molecular profiling can distinguish paediatric and adult acute myeloid leukaemias

6. Methods section: lines 419-421: Murine Stem Cell Virus, "MSCV" not "MSVC"?

Our response:

This has been corrected.

REVIEWERS' COMMENTS:

Reviewer #1 (Remarks to the Author):

Authors have addressed all concerns adequately. Accept revised manuscript.

Reviewer #2 (Remarks to the Author):

The authors have addressed my comments, and the MS has significantly improved. There are a few minor points that they may consider to change to further improve the MS.

1. Fig. 1C & D: it is difficult to appreciate from 1 D that the observed expression levels were not significantly different: if the data from 1C/D is only from a single experiment, then I would not show 1D. I think even if there are some differences, it does not "kill" the story.
2. Fig. 4: I would rearrange the panels A-B-C-...H in a clock-wise manner. and D and E would need to moved down a bit.
3. Fig. 5C: ...does the TCGA dataset contain paediatric samples? I don't think so...in other words, although interesting, the data here indicates that the most promising genes/markers that come out from this comparison of different age-related leukemia....are characterizing adult leukemia! I leave this to the authors...but I find it not very logic to present it this way. so maybe I would first show the link to paediatric AML (Fig.5 L-M-N) and finally talk about the adult AML...indicating a wider function.

Response to reviewers comments:

Reviewer #1 (Remarks to the Author):

Authors have addressed all concerns adequately. Accept revised manuscript.

Our response: Thank you

Reviewer #2 (Remarks to the Author):

The authors have addressed my comments, and the MS has significantly improved. There are a few minor points that they may consider to change to further improve the MS.

1. Fig. 1C & D: it is difficult to appreciate from 1 D that the observed expression levels were not significantly different: if the data from 1C/D is only from a single experiment, then I would not show 1D. I think even if there are some differences, it does not "kill" the story.

Our response: We have removed figure 1D.

2. Fig. 4: I would rearrange the panels A-B-C-..H in a clock-wise manner. and D and E would need to moved down a bit.

Our response: We have rearranged the figure as suggested.

3. Fig. 5C: ...does the TCGA dataset contain paediatric samples? I don't think so...in other words, although interesting, the data here indicates that the most promising genes/markers that come out from this comparison of different age-related leukemia....are characterizing adult leukemia! I leave this to the authors...but I find it not very logic to present it this way. so maybe I would first show the link to paediatric AML (Fig.5 L-M-N) and finally talk about the adult AML...indicating a wider function.

Our response: Yes the TCGA is from adult only but a similar dataset is not publically available for paediatric patients so we were limited in this analysis. We have reordered the presentation of the figures as suggested by the reviewer.